# Competencies of lower-level community health centre leaders in annual health work planning and their influence on district performance in Busoga sub-region: A retrospective study

Kharim Mwebaza Muluya[1,2,3]*, Gangu David Muwanguzi[3], Abdulmujeeb Babatunde Aremu[2], Naziru Rashid[2], Irene Wananda[4], Jonah Fred Kayemba[3], Collin Ogara[3], Musa Waibi[3], John Francis Mugisha[5], Peter Waiswa[1,6]

1 Department of Research and Innovations, Busoga Health Forum, Jinja, Uganda, 2 Department of Public Health, Faculty of Health Sciences, Islamic University in Uganda, Mbale, Uganda, 3 Iganga District Local Government, Iganga, Uganda, 4 Kamuli District Local Government, Kamuli, Uganda, 5 Bishop Stuart University, Mbarara, Uganda, 6 School of Public Health, Makerere University, Kampala, Uganda

* muluyak@gmail.com

## Abstract

### Introduction

Lower-level community health centres play a crucial role in the delivery of primary healthcare services, and the competencies of their leaders can significantly influence district performance. Annual health work planning in local governments faces implementation obstacles every year. This mostly affects lower-level community health centres in Busoga region. It is evidenced by late submission of annual health work plans to authorized offices and also these work plans are poorly made by lower-level community health centres in Busoga region. This prompted a retrospective study to understand the competencies of leaders in the lower-level community health centres in annual health work plan development.

### Objective

The study assessed the influence of competencies of lower-level community health centre leaders in annual health work planning on the district performance in Busoga region.

### Methods

A retrospective (case-control) study design was employed to understand health centre performance across various districts in Busoga sub-region. There was a comparison of performance between the worst performing (case) and best performing (control) districts in the region according to the Annual Health Sector Performance reports from 2017/18 financial year to 2021/2022. Leaders in the lower-level community health centres were

**Data availability statement:** All relevant data are within the manuscript and its Supporting Information files.

**Funding:** The author(s) received no specific funding for this work.

**Competing interests:** Authors declare no competing interests.

**Abbreviations:** AWP, Annual Work Plans; LLGs, Lower Local Governments; NA, National Average; FY, Financial Year; DHIS2, District Health Information System 2; HC, Health Centre; HUMC, Health Unit Management; CVI, Committee Content Validity Index; CI, Confidence Interval; IRB, Institutional Review Boards; MoFPED, Ministry of Finance, Planning and Economic Development

recruited to participate in the study. Data was collected between 17th July, 2024 and 23rd August, 2024. Statistical analysis was conducted on data from 12 case health centres and 12 control health centres using STATA *version 16* to determine competencies of lower-level community health centre leaders that influence district performance.

## Results

The study found that the district performance in annual health work planning was poor in both the case and control groups (26.4% and 47.2% respectively). Only three competencies variables of lower-level community health centre leaders were significantly influencing the performance of the districts. Districts with health facilities which reported that their Health Unit Management Committees (HUMCs) were fully constituted as guided by their leaders for annual health work planning, had significantly higher odds (AOR = 13.551, 95% CI: 4.816–38.617, p < 0.001) and performed better than those whose HUMCs were not fully constituted. Additionally, districts with health facilities which indicated that Heads of Departments (HODs) were involved in the annual health work planning process had significantly higher odds (AOR = 6.500, 95% CI: 3.109–10.791, p = 0.007) of better planning outcomes. Furthermore, districts with facilities that organized annual planning meetings by their leaders had significantly increased odds (AOR = 3.060, 95% CI: 1.399–6.916, p = 0.002) of achieving effective planning and the performance of the district.

## Conclusion

The competencies of lower-level community health centre leaders in fully constituting the health unit management committees, involving heads of departments in the annual planning processes and organizing annual meetings for effective planning, highly contributes to the general performance of districts in the Busoga sub-region. Strengthening these competencies through targeted supportive meetings, consistently appointing HUMC members in health centres and creation of active departments of health centres as initiatives are recommended to enhance the overall effectiveness of health service delivery in Busoga sub-region.

## Introduction

Health care in Uganda is delivered through a decentralised framework with the district responsible for all structures within its confines except referral hospitals where they exist [1]. The framework is organized in such a way that below the National Referral hospitals are the regional referral hospitals, district general hospitals, Health Sub District Health Centres (Health centre IVs), Sub County level Health Centre IIIs, Parish level Health Centre IIs and the Village Health Teams (VHTs). The lower-level community health centres are found in the Lower Local Governments (LLGs) in districts and include health centres IIs, IIIs and IVs [1].

Compilation of an Annual Health Work Plan is an annual recurring activity performed by Health Centre IIs, Health Centre IIIs, and other upper-level government health facilities. These plans provide suitable performance improvement score guides and means of verification according to the Local Government Management of Service Delivery Performance Assessment Manual provided for the higher local governments [2]. Performance improvement is driven by addressing health centre performance gaps to provide good services to the needy population.

Empowering lower-level health centres requires effective coordination and management by competent leaders at different levels including national, regional, district, health sub district, sub-county, parish/ward and village/cell levels [3]. It is recommended that both leaders from the health centres and communities work together in developing annual health work plans [4]. However, it is practically observed in most countries Uganda inclusive that communities and other key stakeholders are not involved in making annual work plan for health centres [3,5].

Countries worldwide, are in the fray of making annual work plans for the better implementation of their programs. Successful countries have achieved it with the help of competent leaders. However, in many countries, weak health systems, lack of coordination, and failed partnerships have been observed and produced lacklustre programmes at different levels of health service delivery due to poor planning [6].

This has not been different for most of the sub-Saharan countries. Health work plans are developed but these countries still experience unique challenges. Ghana discovered that human resource shortages, communication skills, financial constraints, and a narrow decision space, limit the ability of health managers to meet tasks, health planning inclusive [7]. Similarly in Tanzania, lack of awareness on the comprehensive council health plan among health facility governing council members, poor communication and information sharing, unstipulated roles and responsibilities of the governing council, lack of management capacity among council leaders, and lack of financial resources hindered the development of better health work plans [8].

Annual Work Plans (AWP) are the annual work programs to be prepared by health centre leaders during each calendar year. These include a program of activities and their performance in the concluding year followed by a forecast of the schedule of activities proposed for inclusion in the following fiscal year [9]. It is a set of successive interconnected activities over a period of one year, which contribute to the same broader aim and are created to oversee planned activities, and expected results [9]. An annual work plan can be as complex or as simple as the organization wants it to be and the policy-making levels set the barometers [9].

Annual health work plans can negatively or positively be associated with performance for the different health indicators [4]. The main aim of annual planning is to design well focused action plans that will be executed by the local government and related ministries [3]. Health Centre Annual Work Plans are derived based on their allocation as routine and capital development budgets. Different health centres have been given opportunities to develop work plans since they experience different needs and priorities which has automatically exacerbated desirable support from funding bodies [4,10].

The annual health work plan is so much well-thought-out to have local government service delivery results including access to services by population, management of investment projects as per the guidelines through better leadership and commitment, resource mobilization and recruitment of staff, amongst others. Like other countries, discussions on services to be included in the work plan range from health education, to environmental determinants of health, to water and sanitation, to mental health and health-seeking behaviour, amongst other [10].

Improved performance may be determined by the performance reports of health centres extracted from the annual work plans prepared and submitted to the district health office and ministry of health by March 31st of every financial year. Importantly, management, monitoring and supervision of the work plans are necessary to account for district executions by the health centres' budgets for routine services and investment projects [2]. Lower-level community health centres smoothly implement their annual work plans when their health workers are highly motivated to perform their tasks and are given appropriate rewards [3].

The current assessment report by the Office of the Prime Minister (OPM) shows that annual health work plans in local governments have faced obstacles to implementation every year [2]. This challenge has been attributed to weaknesses in the planning

processes at the facility and district levels. As a result, many health facilities in Uganda produce poorly developed annual health work plans [9]. Thus, a gap remains among leaders of lower-level health facilities in making effective and actionable work plans. This is leading to poor health service delivery and causing slow progress to the achievement of the Sustainable Development Goals (SDGs). One of the obstacles to implementation of the work plan is that leaders in the lower-level community health centres are not motivated, satisfied and skilled which affects their health centre performance in the annual work planning process [3]. Similarly, government grants sent to local governments indicate an inconsistency between lower health facilities proposed work plans and the total program delivery priorities funded by the district budget desk. Several of these proposed work plans consistently continue to differ from the priority areas and budget allocations funded in the final consolidated districts' budgets. Health centre leaders' competence in health work planning is not being observed to give added value, particularly in supporting individual health centre and district performance [3]. Therefore, this study was to determine the competencies of lower-level community health centre leaders in annual health work planning and their influence on district performance in the Busoga sub-region.

### Research questions

1) What is the level of performance of districts in annual health work planning in Busoga sub-region?

2) What competencies of lower-level community health centre leaders in annual work planning influence district performance in Busoga region?

### Objectives of the study

**Broad objective.**  The broad objective of the study was to explore the competencies of lower-level community health centre leaders in annual health work planning that influence the districts' performance in the Busoga sub-region.

### Specific objectives

1) To assess the level of performance of districts in annual health work planning in Busoga region.

2) To assess the competencies of lower-level community health centre leaders in annual health work planning that influence district performance in Busoga region.

## Materials and methods

### Study design

This was a retrospective case-control study conducted between July 2017 and June 2022. A retrospective case-control study design was used to empirically assess and explain the significance of leadership competencies in annual health work planning on district performance in the Busoga sub-region. The study categorized districts in the region as poor and good performing based on their annual performance reports from the Ministry of Health over the last five years. Districts which met the national average score were good performing districts. Case districts were those that failed to meet the national average score at least three times within five years, while control districts were those that met or exceeded the national average score at least four times [11–15].

### Study setting

The study was conducted in the Busoga sub-region, which comprises eleven (11) districts and one (1) city. These districts include Mayuge, Namayingo, Bugiri, Bugweri, Namutumba, Kaliro, Luuka, Kamuli, Buyende, Iganga, and Jinja District Local Government, representing the rural districts, while Jinja City is the only urban district in the Busoga sub-region.

## Variables and their measurements

The dependent variable is district performance in annual health work plan development and the independent variable is the competencies of lower-level community health centre leaders in annual health work plan development for the different districts in the Busoga sub-region.

District performance was assessed based on the performance of health facilities in annual health work plan development and its implementation. The annual health work plan has specific areas of focus, which were considered critical indicators of a facility's performance and its overall contribution to district health targets. The focus was on whether a facility had planned and implemented maternal and child health activities, health promotion, disease prevention, sanitation and hygiene activities, administrative activities, infrastructure activities, and routine mobilization activities. Additionally, routine performance reviews and implementation reports were considered as indicators of continuous improvement. District performance was considered good when at least 75% of the priority areas identified in the facility work plan were realized, adopted, and included in the district's final work plan. Each correct activity was scored 1, otherwise 0.

**a) District performance in annual health work plan development.** Performance is typically assessed in terms of outcomes. Similarly, it can be evaluated based on health-seeking behaviour [5]. In Kenya they agreed that performance indicators range from health education, to environmental determinants of health, to water and sanitation, to mental health and health-seeking behaviour, amongst others [10]. One of the studies, stated that employee's performance is measured against the performance standards set by the organization [16]. In Uganda, the ministry of health has set performance standards in line with maternal child health, health education and promotion, environmental determinants of health, water and sanitation, mental health and health-seeking behaviour, amongst others for the lower-level health centre leaders.

There are a number of measures that can be taken into consideration when measuring performance for example using productivity, efficiency, effectiveness, quality and profitability measures [17]. Based on such measures, the district performance was measured as a good one when three quarters (75%) of the priority areas identified in the facility work plan were realized, adopted and included in the final work plan of the district, there is an attached budget to every item and the level of funding in the work plan is realized. This is not far from the annual health sector performance reports for the selected five years where the national average score ranged from approximately 65–74 percent (approx. $3/4$ of the performance mark). Therefore, an average score for the priority areas was considered and a district which scored 75% and above was taken to be a good performing district.

**b) Competence in annual health work plan development.** In this study, competence is defined as the ability of lower-level health centre leaders to successfully and effectively develop an annual health work plan. The study established the influence of competencies of the lower-level health centres' leadership in annual health work planning on total program delivery in Busoga sub-region. Competence requires knowledge and skills among lower-level health centre leaders, the availability of a planning team, active involvement of all stakeholders at the health centres, timely planning meetings, and adherence to planning guidelines, among other factors [10].

In this study, competencies of lower-level health centre leaders were defined by their ability to engage community stakeholders like the mandated HUMC to participate in annual health work planning. Equally, managerial and administrative competencies were also considered. For instance, leaders should exercise the skills of using the limited staff for results (work planning) through motivation, close supervision or invitations to participate in the annual work planning. Other key competencies include problem-solving and decision-making skills, enabling leaders to be innovative during planning meetings, as well as technical and professional competencies, which allow health centre leaders to identify and understand national policies and guidelines used in annual health work planning.

Therefore, qualified and motivated health centre leaders are essential for adequate health service provision, however, it is still a challenge in rural areas. Yet, according to the World Health Organisation (WHO), performance especially in rural settings is considered to be a combination of leaders' competencies, productivity and responsiveness [18]. There is

statistically significant evidence of health work plans being useful in the implementation of activities generated by competent health workers and leaders in communities [19].

## Selection criteria of districts

The study considered the following;

1. The district must have existed throughout the study period.

2. Districts that performed below the national average (NA) for at least three years between 2017/18 and 2021/22 were classified as case districts (poor-performing districts).

3. Districts that performed above the national average (NA) for at least four years between 2017/18 and 2021/22 were classified as control districts (good-performing districts).

4. Only rural districts were considered for this study, and Jinja City was excluded.

As shown in Table 1, Namutumba, Namayingo, Mayuge, and Luuka districts were selected as case districts, while Jinja, Iganga, Kamuli, and Kaliro were chosen as control districts.

## Sampling techniques

Purposive sampling was used to select the districts. This method is designed to enhance the understanding of specific individuals' or groups' experiences and to aid in developing theories or concepts. Researchers achieve this by selecting "information-rich" cases—individuals, groups, organizations, or behaviours that provide valuable insights [20].

A stratified random sampling approach was used to select lower-level public Health Centre IVs, IIIs, and IIs in the Busoga sub-region. A multistage stratified sampling method was applied to determine the number of health centres (HCIVs, HCIIIs, and HCIIs) selected from both case and control districts for data collection.

## Study sample size determination

According to the District Health Information System 2 (DHIS2), the total number of HCIVs, HCIIIs, and HCIIs in each district and city in the Busoga region is presented in Table 2 [21]. In total, 330 health centres were considered as the sampling frame for this study.

Table 1. Extract of district league table performance for the period 2017/18 to 2021/22.

| District | FY 2017/18 NA (69.2) | FY 2018/19 NA (73.1) | FY 2019/20 NA (68.1) | FY 2020/21 NA (64.5) | FY 2021/22 NA (68) |
|---|---|---|---|---|---|
| Bugiri | 73 | 70.7 | 72.09 | 64.7 | 60.9 |
| Bugweri | – | 64.5 | 72.74 | 53.3 | 58.8 |
| Buyende | 65.8 | 78.7 | 74.04 | 60.6 | 70.1 |
| Iganga | 72.1 | 74.9 | 73.53 | 66.8 | 67.2 |
| Jinja city | – | – | – | – | – |
| Jinja | 79.5 | 85 | 80.35 | 66.7 | 64.6 |
| Kaliro | 71 | 66.3 | 71.36 | 70.1 | 68 |
| Kamuli | 70.9 | 74.6 | 71.96 | 66.1 | 65.3 |
| Luuka | 56.3 | 68.7 | 65.94 | 63.1 | 65.9 |
| Mayuge | 63.7 | 66.3 | 70.58 | 55.7 | 58.7 |
| Namayingo | 66.2 | 70.7 | 70.21 | 60.9 | 66.7 |
| Namutumba | 73.7 | 72.4 | 59.29 | 60.2 | 61.5 |

Source: Annual health sector performance reports; 2017/18, 2018/19, 2019/20, 2020/21 and 2021/2022

Muwanguzi and others conducted a study on the effectiveness of training for Health Unit Management Committee (HUMC) members, citing Schelssman's (1982) formula to determine the required sample size for health centres [5].

$$n = \frac{(1 + 1/c)\, P_1 Q_1 \,(Z_\alpha + Z_\beta)^2}{(P_1 - P_0)^2}$$

Where:

n = required sample size of the case group
$Z_\alpha$ = standard normal value corresponding to a significance level of 0.05 (1.96)
$Z_\beta$ = standard normal value corresponding to 80% study power (0.84)
$P_0$ = probability of success among control districts
$P_1$ = probability of success among case districts
C = number of controls per case (C = 1)
$Q = 1 - P_1$

A study by Ibrahim and others on implementing grip strength measurement in medicine for older patients in Newcastle, UK, identified $P_0$ as 46.3% and $P_1$ as 79% [22]. For this study, 12 health centres were selected from the case districts and 12 from the control districts. The health centres in each district category (good or poor-performing) were randomly chosen for data collection. Table 3 provides details on the number of health centres selected per district.

### Data collection techniques

**a) Documentary review.** Under the secondary collection of data, documented records provide the necessary background and a lot of necessary information for a more organized work [23]. Where possible, depending on the cooperation of district managers and health centre leaders, the researcher reviewed the annual health work plans at the health centres to understand the work plan processes, its submission, its contents and adoption by the district budget desk, performance management records and others. The data collection process took place between 17th July, 2024 and 23rd August, 2024. The review concentrated much on maternal child health indicators, health education and promotion,

**Table 2. Number of health centre IVs, IIIs & IIs for districts in Busoga sub region.**

| Districts | Level of Health Centre | | | |
|---|---|---|---|---|
| | IV | III | II | Overall total |
| Bugiri | 1 | 10 | 26 | 37 |
| Bugweri | 1 | 3 | 12 | 16 |
| Buyende | 1 | 4 | 12 | 17 |
| Iganga | 1 | 9 | 18 | 28 |
| Jinja city | 4 | 8 | 13 | 27 |
| Jinja | 1 | 7 | 24 | 32 |
| Kaliro | 1 | 6 | 7 | 14 |
| Kamuli | 2 | 12 | 26 | 40 |
| Luuka | 1 | 6 | 22 | 29 |
| Mayuge | 3 | 9 | 28 | 40 |
| Namayingo | 1 | 5 | 21 | 27 |
| Namutumba | 1 | 4 | 18 | 23 |
| Total | 18 | 83 | 227 | 330 |

Source: DHIS2 21 March, 2023

**Table 3. Number of health centre IVs, IIIs & IIs selected for the study in the four districts.**

| Performance of districts | | Level of Health Centre | | | |
|---|---|---|---|---|---|
| | | IV | III | II | Total |
| Poor performance | Overall number of HCs | 5 | 34 | 85 | 124 |
| | Sampled number of HCs | 1 | 3 | 8 | 12 |
| Good performance | Overall number of HCs | 6 | 24 | 89 | 119 |
| | Sampled number of HCs | 1 | 3 | 8 | 12 |

water and sanitation and other environmental health related indicators, mental health and health-seeking behaviour, infrastructure, amongst others.

**b) Structured interviews.** Leaders were asked questions to gain information on their competencies as supported by Amin [24]. This was a questionnaire with different sections of questions answered by leaders on interview by research assistants at the health centres. Appointments with lower-level community health centre leaders were made to avoid missing the leaders at the health centres visited.

## Research instruments

Clifton and Handy stated that choosing among the different data collection tools involves considering their appropriateness and relative strengths and weaknesses [25]. In this study, combinations of tools were used, that is, records review checklist and questionnaire. These tools were designed using the key study themes/ objectives. The secondary data sources included work plans for the last five years at health centres and the annual assessment service delivery performance reports at the districts.

**a) Questionnaire.** A questionnaire is a set of written questions developed by the researcher based on the study objectives and literature review and administered to a selected group of respondents [25]. It is particularly useful when dealing with respondents who have limited time for interviews and feel more comfortable expressing their views in writing.

For this study, a standardized document containing pre-formulated questions—primarily closed-ended, with a few multiple-choice questions—was used to collect responses from consenting lower-level community health centre leaders. Research assistants administered the questionnaires and recorded responses from participants across different health centres.

**b) Document/Records review checklist.** A checklist is a tool designed to capture data extracted from secondary sources. In this study, it was used to verify specific responses provided by participants regarding annual health work plans and annual service delivery performance assessment reports at lower-level health centres in the Busoga region over the past five years.

## Data quality control

**a) Validity of Instruments.** Validity involves getting the most accurate data. It is defined as the degree to which a test or measuring instrument, measures what it purports to measure, or how well a test or an instrument fulfils its function [26]. Content Validity Index (CVI) was used to get the validity of the research instruments and it was performed based on items derived from instruments and volunteer evaluators. Each evaluator rated the questions on a two-point rating scale of *Relevant (R)* and *Irrelevant (IR)*. Thereafter, the computation of CVI was done by summing up the judges' ratings on either side of the scale and dividing the two to obtain the average.

**b) Reliability of instruments.** The researcher pre-tested the instruments to ascertain the level of consistency, weakness, and unclear questions to 10 respondents and adjustments were done to enhance its reliability. The researcher also used Cronbach's alpha coefficient on analysis to identify the variables with an alpha coefficient of more than 0.70, which is acceptable for social research [24].

### Data analysis

Quantitative data were analyzed using STATA version 16. Tests of independence were conducted to determine the statistical significance of different variables, with a p-value set at 0.05 (95% confidence interval) to assess the influence of independent variables on the dependent variable.

A logistic regression model was used to evaluate the statistical relationship between independent and dependent variables. Additionally, odds ratios and p-values were analyzed to determine key factors influencing the study outcomes. Furthermore, multiple logistic regression was performed to confirm whether independent variables were significant predictors of performance improvement.

### Ethical approval and consent to participants

Ethical approval to conduct the study was provided by the Institutional Review Boards (IRB) of Uganda Martyrs University – Nsambya hospital (SFHN – 2024–134). Written consent was obtained from all the study participants and authorities in the study area. Privacy and confidentiality of the information was assured to the respondents. The participants consented to participate in the study and were assured of their right to withdraw from the engagement at their will. This is a key aspect of the consenting process for the study participants.

## Results

A total of 24 health centres participated in the study, with 12 selected from poor-performing districts (cases) and 12 from well-performing districts (controls). These included two HCIVs (one from a case district and one from a control district), six HCIIIs (three from case districts and three from control districts), and 16 HCIIs (eight from case districts and eight from control districts). The study covered five financial years, yielding 120 possible observations—60 from the case districts and 60 from the control districts.

### Districts' performance in annual health work plan development in Busoga sub-region

Overall, the performance of districts in annual health work plan development was poor in both cases and controls, at 26.4% and 47.2%, respectively. This performance was influenced by the effectiveness of health centers in planning and implementing annual health work plans, particularly in areas such as budget allocation and execution rate, prioritization of key health services (notably maternal and child health), water, sanitation, hygiene, and health promotion.

Districts in the control group performed slightly better than those in the case group as seen in Fig 1. However, this level of performance was still below the desired target of 75%, both for districts and health centres.

### Competencies of leaders of lower-level community health centres in annual health work plan development

In this study, the competence of lower-level health centre leaders in engaging community stakeholders in health centre management was considered crucial. Health Unit Management Committee (HUMC) members, as key community stakeholders, are mandated to participate in the development of the annual health work plan. Therefore, each health centre should have a formally constituted HUMC.

As shown in Table 4, lower-level health centre leaders had initiated the process, and HUMCs were constituted in all health centers (both case and control) at 100%. Interestingly, 90% of leaders in the control districts had fully constituted their HUMCs, compared to 50% in the case districts. Additionally, about three-quarters of all health centres in both case and control districts had HUMC members who met the academic requirements set by the Ministry of Health in Uganda.

For managerial and administrative competencies, the number of staff a leader supervised, motivated, or invited to participate in annual work planning was a key factor. This involvement enabled leaders to demonstrate their ability

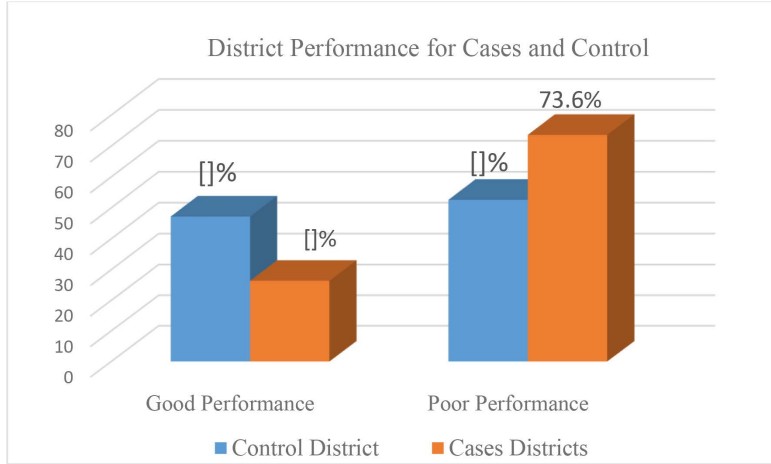

**Fig 1. Overall performance of cases and control districts.**

to assign tasks during the annual health work planning period. However, only 13.3% of health centres in the control districts and 10% in the case districts met the required staffing levels. More details on competence activities are provided in Table 4.

Problem-solving and decision-making competencies were evident when leaders demonstrated innovation in improving service delivery. Planning meetings provided an opportunity to discuss innovative strategies. However, approximately 80% of health centre leaders in the control districts and 83.3% in the case districts had not attended four or more planning cycle meetings organized by the district over five years. Similarly, most health centre leaders lacked the competence to organize annual health work planning meetings at the facility level and to meet the required quorum for such meetings. In many cases, fewer than four people were involved in the planning process for developing the annual health work plan (43.3% in control districts and 70% in case districts).

Technical and professional competencies were essential for health centre leaders to identify and understand national policies and guidelines used in annual health work planning. Several tools are available to guide the planning process; however, only 13.3% of health center leaders in both control and case districts were able to use at least three or more tools for effective planning (see Table 4).

**Competencies of the lower-level health centre leaders influencing districts' performance in the Busoga sub-region**

The following competency variables significantly influenced district performance (p-value < 0.05); the ability of leaders to fully constitute Health Unit Management Committees (HUMCs), the ability to discuss innovations in planning cycle meetings organized by district health departments, the ability to mobilize Heads of Departments (HODs) to attend annual planning meetings, and the ability to organize meetings at the health facility.

As seen in Table 5, districts with facilities where leaders had fully constituted HUMCs for annual health work planning were 14 times more likely to perform well compared to those without a fully constituted HUMC (COR = 14.000, 95% CI: 4.516–43.431, p < 0.001).

Similarly, the involvement of HODs in the annual planning process was significantly associated with better facility performance. Districts with facilities where HODs participated in planning were 7 times more likely to perform well compared to those where they were not involved (COR = 7.000, 95% CI: 2.99–16.40, p = 0.005).

**Table 4.  Competencies of lower-level community health centres in Busoga region.**

| Variables/Activities | Control | Case |
|---|---|---|
| | Freq. (%) | Freq. (%) |
| Ability to engage community stakeholders (HUMCs). There is a HUMC constituted with guidance of health centre leaders on the process. | | |
| Yes | 60 (100) | 60 (100) |
| No | 0 (0.0) | 0 (0.0) |
| HUMC fully constituted as per the guidelines by leaders | | |
| Yes | 56 (90) | 30 (50.0) |
| No | 4 (10) | 30 (50.0) |
| HUMC members meeting the academic requirements | | |
| Yes | 45 (75) | 46 (76.7) |
| No | 15 (25) | 14 (23.3) |
| For the managerial and administrative competencies, is the staffing level meeting facility standards for the leader to exercise the ability of assigning tasks to individuals during planning period | | |
| Yes | 8 (13.3) | 6 (10) |
| No | 52 (87.7) | 54 (90) |
| Ever attended planning cycle meetings organized by the district health department for innovative discussions. | | |
| Yes | 12 (20) | 10 (16.7) |
| No | 48 (80) | 50 (83.3) |
| Number of planning cycle meetings organized by the district health department you attended. | | |
| Attended at least 4 times for the 5 years | 12 (20) | 10 (16.7) |
| Attended less than 4 times for the 5 years | 48 (80) | 50 (83.3) |
| Organized annual planning meetings at the facility | | |
| Yes | 34 (56.7) | 18 (30) |
| No | 26 (43.3) | 42 (70) |
| Ability to involve people in the annual planning process (HODs, HUMCs etc.). | | |
| 4 and more persons involved | 34(56.7) | 18 (30) |
| < 4 persons involved | 26(43.3) | 42 (70) |
| Ability to identify and interpret tools used for developing annual health work plan: | | |
| 3 and more tools mentioned | 8 (13.3) | 8 (13.3) |
| < 3 tool mentioned | 52 (86.7) | 52 (86.7) |

Organizing annual planning meetings at the facility level also showed a significant association with performance. Districts with facilities that conducted these meetings were 3.05 times more likely to perform well than those that did not (COR = 3.051, 95% CI: 1.435–6.471, p = 0.001). Additionally, attendance at planning cycle meetings organized by the district health department was statistically significant in relation to performance (COR = 1.250, 95% CI: 0.493–3.168, p = 0.001). See Table 5.

## Predictors of district performance on multiple regression analysis

Districts with health facilities whose leaders reported that their Health Unit Management Committees (HUMCs) were fully constituted for annual health work planning, had significantly higher odds (AOR = 13.551, 95%

**Table 5. Competence of the lower-level health centre leaders in annual health work planning influencing performance of districts.**

| Variables | Control (Good performance) Frequency (%) | Case (Poor performance) Frequency (%) | COR (95% CI) | p-value |
|---|---|---|---|---|
| Ability of leaders to constitute a HUMC | | | | |
| Yes | 50 (83.3) | 40 (66.7) | 0.812 (0.296-2.008) | 0.118 |
| No | 10 (16.7) | 20 (33.3) | 1.0 | |
| HUMC fully constituted by leaders | | | | |
| Yes | 56(65.1) | 30(34.9) | 14.0 (4.516-43.431) | **0.001** |
| No | 4(11.8) | 30(88.2) | 1.0 | |
| Ability to identify HUMC members meeting the academic requirements | | | | |
| Yes | 45(49.5) | 46(50.5) | 0.913(0.396-2.106) | 0.136 |
| No | 15(51.7) | 14(48.3) | 1.0 | |
| Staffing level meeting the standards for the facility for the leader to exercise the ability of assigning tasks to individuals during planning period | | | | |
| Yes | 8(57.1) | 6(42.9) | 1.385(0.451- 4.251) | 0.663 |
| No | 52(49.1) | 54(50.9) | 1.0 | |
| Leaders involve HODs in the annual planning process | | | | |
| Yes | 35(77.8) | 10(22.2) | 7.000 (2.99-16.40) | **0.005** |
| No | 25(33.6) | 50(66.7) | 1.0 | |
| Leaders involve HUMCs in the annual planning process | | | | |
| Yes | 30(54.5) | 25(45.5) | 1.400(0.682-2.87) | 0.085 |
| No | 30(46.2) | 35(53.8) | 1.0 | |
| Leaders ever attended planning cycle meetings organized by the district health department | | | | |
| Yes | 12(54.5) | 10(45.5) | 1.250 (0.493-3.168) | **0.001** |
| No | 48(49.0) | 50(51.0) | 1.0 | |
| Leader organize annual planning meetings at the facility | | | | |
| Yes | 34(65.4) | 18(34.6) | 3.051(1.435-6.471) | **0.001** |
| No | 26(38.2) | 42(61.8) | 1.0 | |
| Leaders able to identify and utilize manual for annual health planning | | | | |
| Yes | 8 (50.0) | 8 (50.0) | 1.000 (0.348-2.868) | 0.389 |
| No | 52(50.0) | 52 (50.0) | 1.0 | |

CI: 4.816–38.617, p < 0.001) and significantly performed well compared to those whose HUMCs were not fully constituted as shown in Table 6.

Additionally, districts with health facilities which indicated that Heads of Departments (HODs) were involved in the annual planning process had significantly higher odds (AOR = 6.500, 95% CI: 3.109–10.791, p = 0.007) of better planning outcomes (district performance). Furthermore, districts with facilities whose leaders organized annual planning meetings had significantly increased odds (AOR = 3.060, 95% CI: 1.399–6.916, p = 0.002) of achieving effective planning and the performance of the district. See Table 6.

## Discussion

The findings of this retrospective study indicate that specific competencies of leaders in lower-level community health centres significantly influence the performance of districts in the Busoga sub-region (see S1 and S2 Data). Notably, three key competencies emerged as significant contributors to district performance.

**Table 6. Multiple regression for predictors performance of districts.**

| Variables | COR (95% CI) | p-value | AOR (95% CI) | p-value |
|---|---|---|---|---|
| Ability of leaders to constitute a HUMC | | | | |
| Yes | 0.812 (0.296-2.008) | 0.118 | 1.100 (0.448-2.968) | 0.387 |
| No | 1.0 | | 1.0 | |
| HUMC fully constituted by leaders | | | | |
| Yes | 14.000 (4.516-43.431) | <0.001* | 13.551 (4.816-38.617) | **<0.001*** |
| No | 1.0 | | 1.0 | |
| Ability to identify HUMC members meeting the academic requirements | | | | |
| Yes | 0.913(0.396- | 0.136 | 0.984 | 0.273 |
| No | 2.106) 1.0 | | (0.587-2.734) 1.0 | |
| Staffing level meeting the standards for the facility for the leader to exercise the ability of assigning tasks to individuals during planning period | | | | |
| Yes | 1.385(0.451- 4.251) | 0.663 | 1.300 (0.450-4.500) | 0.301 |
| No | 1.0 | | 1.0 | |
| Leaders involve HODs in the annual planning process | | | | |
| Yes | 7.000 (2.99-16.40) | 0.005* | 6.500 (3.109-10.791) | **0.007*** |
| No | 1.0 | | | |
| Leaders involve HUMCs in the annual planning process | | | | |
| Yes | 1.400(0.682-2.87) | 0.085 | 1.690 (0.600-2.800) | 0.120 |
| No | 1.0 | | 1.0 | |
| Leaders ever attended planning cycle meetings organized by the district health department | | | | |
| Yes | 1.250 (0.493-3.168) | <0.001 | 1.671 (0.428-3.635) | 0.150 |
| No | 1.0 | | 1.0 | |
| Leader organize annual planning meetings at the facility | | | | |
| Yes | 3.051(1.435-6.471) | 0.001 | 3.060 (1.399-6.916) | **0.002*** |
| No | 1.0 | | 1.0 | |
| Leaders able to identify and utilize manual for annual health planning | | | | |
| Yes | 1.000 (0.348-2.868) | 0.389 | 1.101 | 0.415 |
| No | 1.0 | | (0.447-2.815) | |

1. Health centre leaders initiated the process of constituting the HUMCs and there was presence of a fully constituted Health Unit Management Committee (HUMC), which is under the mandate of ministry of health. The HUMC are leaders who actively participate in decision-making, and can significantly impact the performance of overall health service delivery. According to the national guidelines, a fully constituted HUMC will include local government representatives, community leaders, health facility in-charges, representatives from Village Health Teams (VHTs) and health workers in the facility who are instrumental in the development of work plans. Studies have shown that a properly constituted HUMC contributes to good performance of districts in terms of governance and accountability, community participation, resource mobilisation, amongst others [6]. Findings of this study were contrary to most of the studies conducted especially in the sub-Saharan Africa [3,5,6,8].

2. Participation in planning cycle meetings is very necessary to health centre leaders for proper planning. Ministry of Finance, Planning and Economic Development (MoFPED) and ministry of health share guidelines each financial year for facility leaders to plan better [27]. Always health centres are invited to the district for planning

meeting and those that actively participate showed a significant improvement in performance. However, the low turn up of leaders of lower community health centres for the planning meetings highly affects performance of districts. Perhaps it is attributed to lack of competent leaders and financial constraints as stated in one of the studies [10].

3. Heads of departments when involved in the planning by facility leaders, contribute to better performance of the district as was revealed in the results of this study. Unlike this study, other countries have community representatives who continue not to attend these meetings. In Kenya for instance, the community perspective was brought by the facility committee, in which each village intended to be represented by a committee member during the planning period in addition to the health centre in-charges [10]. Involvement in these meetings ensures that local health centres are aligned with district-wide health priorities, resource allocations, and implementation strategies. Like in Kenya, the committees discussed their inputs in the work plan and the barriers to health at each life stage based on the guidelines which were issued by the MoH [10]. Similarly, it can be encouraged to routinely invite HODs to participate in the planning, since this study deems it important. This alignment is crucial for the effective coordination of healthcare delivery, enabling lower-level centres to contribute meaningfully to district-wide health improvement initiatives.

The combination of these competencies of lower-level health centre leadership, particularly in the areas of fully constituted HUMC, participation in district planning processes, and involvement of heads of departments, plays a pivotal role in improving district health performance. It highlights the need for continuous capacity building for health centre leaders to ensure they have the skills and resources necessary to contribute effectively to district health objectives.

## Conclusion

This study underscores the critical role of lower-level community health centre leaders in driving districts health performance improvement in the Busoga sub-region. The competencies of lower-level community health centre leaders in initiating the process and fully constituting the health unit management committees, involving heads of departments in the annual planning processes and organizing annual meetings for effective planning, highly contributes to the general performance of districts in the Busoga sub-region. These findings suggest that investing in leadership competencies at the community health centre level is essential for achieving district health performance goals. Additionally, the study provides empirical evidence that strengthening these key competencies can lead to statistically significant improvements in health outcomes at the health centre and district levels.

## Recommendations

1. Ensure consistent appointment of HUMC members. HUMC members should be consistently appointed and maintained to provide strong leadership and oversight, ensuring quality standards, efficient planning, and effective management at health centers.

2. Encourage health center leaders to establish active departments. Establishing competent and well-structured departments within health centers enhances organizational efficiency and promotes focused planning and management of health services.

3. Strengthen leadership competencies through targeted supportive meetings. Regular supportive meetings and capacity-building sessions should be organized to provide guidance, problem-solving opportunities, and continuous professional development for health center leaders. These initiatives will ensure continuous improvement in leadership competencies and the ability to address specific challenges effectively.

## Supporting information

**S1 Appendix. Questionnaire on Competencies of Lower-Level Health Centre Leaders and Performance of districts.** This appendix presents the structured questionnaire used to assess the managerial, technical, and interpersonal competencies of leaders at lower-level community health centres. The tool also captures key performance areas and its adaptation during budgeting and final allocations for implementation.
(DOCX)

**S1 Data. Adjusted Data on key focus areas of performance.** This dataset contains cleaned and adjusted responses from the competency assessment of lower-level community health centre leaders, along with corresponding district-level performance indicators. Variables include leadership competencies, health service delivery outcomes, supervision frequency, and resource utilization metrics.
(XLSX)

**S2 Data. Adjusted Data on overall Performance.** This dataset includes adjusted quantitative data examining the relationship between the competencies of lower-level community health centre leaders and overall district health system performance.
(XLSX)

## Acknowledgments

The authors extend their sincere appreciation to Paul Kitakule, Mohammed Mukalu, District Health Officers, Biostatisticians and In-charges of health centres for their support in data collection. We are also grateful to the study participants, lower community leadership and the centre staff in the study region for their contributions to this study.

## Author contributions

**Conceptualization:** Kharim Mwebaza Muluya, John Francis Mugisha, Peter Waiswa.

**Data curation:** Kharim Mwebaza Muluya, Gangu David Muwanguzi, Areemu Babantunde, Naziru Rashid, Irene Wananda, Jonah Fred Kayemba, Musa Waibi.

**Formal analysis:** Kharim Mwebaza Muluya, Areemu Babantunde, Naziru Rashid, Jonah Fred Kayemba, Musa Waibi, John Francis Mugisha.

**Investigation:** Kharim Mwebaza Muluya, Gangu David Muwanguzi, Jonah Fred Kayemba.

**Methodology:** Kharim Mwebaza Muluya, Gangu David Muwanguzi, Collin Ogara, Peter Waiswa.

**Writing – original draft:** Areemu Babantunde, Collin Ogara.

**Writing – review & editing:** Areemu Babantunde, Collin Ogara, Peter Waiswa.

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
