## [Decision Letter · Decision Letter 0]

Dear Dr. Muluya,

Thank you for submitting your manuscript to PLOS ONE. After careful consideration, we feel that it has merit but does not fully meet PLOS ONE’s publication criteria as it currently stands. Therefore, we invite you to submit a revised version of the manuscript that addresses the points raised during the review process.

**As you address the reviewer comments, please explain clearly the case control design and define clearly what your cases were and the controls. **

We look forward to receiving your revised manuscript.

Kind regards,

Aloysius Gonzaga Mubuuke

Academic Editor

PLOS ONE

**Journal Requirements:**

2. In the online submission form, you indicated that data will be available on request.

4. Please include your tables as part of your main manuscript and remove the individual files. Please note that supplementary tables (should remain/ be uploaded) as separate "supporting information" files.

5. We note that this data set consists of interview transcripts. Can you please confirm that all participants gave consent for interview transcript to be published?

If they DID provide consent for these transcripts to be published, please also confirm that the transcripts do not contain any potentially identifying information (or let us know if the participants consented to having their personal details published and made publicly available). We consider the following details to be identifying information:

- Names, nicknames, and initials

- Age more specific than round numbers

- GPS coordinates, physical addresses, IP addresses, email addresses

- Information in small sample sizes (e.g. 40 students from X class in X year at X university)

- Specific dates (e.g. visit dates, interview dates)

- ID numbers

Or, if the participants DID NOT provide consent for these transcripts to be published:

- Provide a de-identified version of the data or excerpts of interview responses

- Provide information regarding how these transcripts can be accessed by researchers who meet the criteria for access to confidential data, including:

a) the grounds for restriction

b) the name of the ethics committee, Institutional Review Board, or third-party organization that is imposing sharing restrictions on the data

c) a non-author, institutional point of contact that is able to field data access queries, in the interest of maintaining long-term data accessibility.

d) Any relevant data set names, URLs, DOIs, etc. that an independent researcher would need in order to request your minimal data set.

For further information on sharing data that contains sensitive participant information, please see: https://journals.plos.org/plosone/s/data-availability#loc-human-research-participant-data-and-other-sensitive-data

If there are ethical, legal, or third-party restrictions upon your dataset, you must provide all of the following details (https://journals.plos.org/plosone/s/data-availability#loc-acceptable-data-access-restrictions):

a. A complete description of the dataset

b. The nature of the restrictions upon the data (ethical, legal, or owned by a third party) and the reasoning behind them

c. The full name of the body imposing the restrictions upon your dataset (ethics committee, institution, data access committee, etc)

d. If the data are owned by a third party, confirmation of whether the authors received any special privileges in accessing the data that other researchers would not have

e. Direct, non-author contact information (preferably email) for the body imposing the restrictions upon the data, to which data access requests can be sent

**Additional Editor Comments:**

Please clarify on the issue of cases and controls especially by clearly defining them.

Reviewers' comments:

Reviewer's Responses to Questions

**Comments to the Author**

1. Is the manuscript technically sound, and do the data support the conclusions?

Reviewer #1: Partly

Reviewer #2: Partly

2. Has the statistical analysis been performed appropriately and rigorously?

Reviewer #1: No

Reviewer #2: No

3. Have the authors made all data underlying the findings in their manuscript fully available?

Reviewer #1: Yes

Reviewer #2: Yes

4. Is the manuscript presented in an intelligible fashion and written in standard English?

Reviewer #1: No

Reviewer #2: Yes

**Reviewer #1:**  REVIEW COMMENTS

1. TITTLE: The word competence being a noun, would sound proper if it was ‘’COMPETENCES IN ANNUAL WORK PLANNING’’ than ‘’COMPETENCE IN WORK PLANS’’.

2. Document/ records review checklist.

Validity of instruments

Reliability of the instruments

Data analysis

Use the right tense (past) and grammar in this section.

3 Dependent variable: The study considers the competences of the leaders of the health centers but in actual sense, it is the competences of the health facilities. The researcher may need to explicitly state the assumption that the health facility competences are synonymous with the leaders’ competences.

4 Methodology: This is a case control study according to the researcher. The controls however do not meet the criteria set out by the study. In fact, the control group participants are cases by definition. There has to be a justification for this decision by the researcher or else this is a major methodological flaw.

It would have been better if the researchers considered a retrospective cohort of under performers( more so after noticing that the control group criteria was out of reach). In this case an independent t-test would have been a good choice at analysis.

5 conclusion The study makes bold conclusions on the backdrop of a methodological error.

6 Readability: The manuscript needs to be trimmed to a more concise version.

OVERALL COMMENT The study presents an extremely relevant and interesting topic. However, it needs major alterations in the methodology for its impact to be realized in the scientific community.

**Reviewer #2:**  Dear Editor,

I hereby submit a review report of the manuscript, entitled ‘Competences of Lower Community Health Centre Leaders in Annual Health Work Plans and Its Influence On District Performance Improvement In Busoga Sub-Region: A Retrospective Study.”. I acknowledge the efforts of the authors for performing a good job and writing evidence which is highly needed. Since, it needs further amendment, I suggested to be accepted with Major revision. I believe, this manuscript will be very helpful in forwarding better evidence for decision making in rebuilding the human resource development and improving the performance of health centers.

Competences of Lower Community Health Centre Leaders In Annual Health Work Plans And Its Influence On District Performance Improvement In Busoga Sub-Region: A Retrospective Study

Title: is too long not catchy. Authors are advised to shorten their title to be attractive for readers.

If authors agree they can use the alternative title that reads as: “Competencies of frontline health facility leaders in annual work plan and its influence on district health management in Busoga Sub-Region: A Retrospective Study”

Abstract

Background: Authors need to incorporate the essential gap that makes them to conduct this research study.

Objective: The first statement is adequate for the objective, while authors need to move the next statement indicated in the objective to the method’s section.

Methods: what is the justification of authors on the adequacy of the sample of health centers for cases and controls? Authors are advised to provide valid justification for the cutoff point “12”.

Authors need to briefly describe the techniques of data collection.

Results: Authors are supposed to describe the number of facilities participated in the study as cases and controls.

In the methods section, Authors noticed: “There was a comparison of performance between the worst performing (case) and best performing (control) districts in the region according to the Annual Health Sector Performance reports from 2017/18 financial year to 2021/2022”. While in their result section they reported “the district performance in annual health work planning was not good in both the case and control groups (26.4% and 47.2% respectively).” How do authors justify for the 1.8 times fold of performance of control groups?

The result showed the 1.8 fold of best performing districts (controls) were with poor performance compared to the worst performing districts. How do authors justify the discrepancy that seems unrealistic?

Conclusion: authors are advised to make conclusion based on their findings.

Keywords: it is better if authors limit the key words to 5

Introduction

The introduction section is not well organized in describing the pertinent information. It is highly dominated by the “Annual work plan concept” which is not aligned with the topic. The problem is not clearly stated. It doesn’t dictate the global experiences and the major identified gaps related to the topic.

The gap is not clearly described in the introduction section. What is the major gap that needs to be explored by authors? Is it about the leadership competency of low health center leaders or organizational annual work plan?

Authors are advised to clearly describe the pertinent components of:

1. The regional (African) experience of low health centers leaders competency in developing annual work plan and its contribution on the district performance improvement

2. The flow of their description needs to be from global to local and from general to specific.

On the second paragraph of the introduction: Authors noticed “In the Uganda Health Sector, the Annual Health Work Plan ………. performed by health centre II, Health Centre III, and ………… provide suitable performance improvement score guides and means of verification …….. for the higher local governments (2020)”. Readers might not easily understand the type of health centers except Ugandan citizens, Authors are advised to briefly describe the health system structure of the country.

Research questions

Authors enquired two research questions: 1) What is the districts’ performance in annual health work planning in the Busoga sub-region?

2) What are the competences in annual health work planning of lower health facilities

leaders that influence the performance of the districts?

However, the research questions formulated by the authors is not in line with their research title.

Authors are advised to rephrase the research questions: In their title authors are interested to explore the competency of low level leaders engaged at health centers and the performance improvement which is directly related to the Annual Work Planning.

Objectives of the study

Specific objectives: the specific objectives are not aligned with the broad objective.

Authors described the specific objectives: 1) To establish the districts' performance in annual work planning in the Busoga sub-region.

2) To establish the influence of competence of the lower community health centre leaders in

annual health work planning on the district performance in the Busoga sub-region.

Authors are advised to align the specific objectives with the general objectives.

Study setting

Authors stated “The study was conducted in the Busoga sub-regions of eleven (11) districts and one (1) city”, what does districts mean? Urban sub-urban or rural districts?

Do the performance assessment standard fits to evaluate Jinga City equally with the other district? Do authors have reasons to include the City with the other district, doesn’t it create a bias to categorize city with other districts in research? For obvious reasons in cities, the competence of health workers and its managerial status is much better compared to the level of districts. Any reflections from the authors.

On another note, authors are advised to provide clear picture of the study setting related to the structure of the health system, availability number of health workers and the access of health institutions to the population.

Authors used District performance and Competence in annual health work planning to measure the variables for performance. It seems many variables are measured in order to determine the status the outcome variable. How did authors manage the analysis of multiple variables?

Sampling technique

Authors noted “Multistage stratified sampling was considered to determine the number of health centre IVs, IIIs and IIs selected for the public facilities in the case and control districts where data was collected.” Authors are advised to provide brief description about the type of health facilities in the study setting section. It is not clear for readers what the meaning of health center IV, III and II. The reason why authors used the stratified sampling technique is not clearly described in the manuscript.

Data collection

Under the Document/Records Review Checklist sub heading, on the data collection section. Authors noted “A checklist will be a tool designed to capture information extracted from the annual health work…….” The “will be” phrase is not acceptable at this stage; authors are advised to us the appropriate form of tense. Similar error is shown on the statement provided under data analysis stated as “The pvalue set at 0.05 will be used to determine the statistical significance ……… at 95 percent confidence intervals (CI)”. “It will derive regression odds ratios and p-values” Authors are advised to make correction on the highlighted phrases. Correction form of the tense is needed.

The sub heading stated by authors “Districts' performance in annual health work planning in Busoga sub-region” and its content in the first paragraph doesn’t fit to the results section. Authors need to consider to move it to the methods section, because it is a descriptive narrative about the performance assessment and how health centers were graded based on their performance.

Similarly, the second paragraph that reads as “Each correct activity was scored 1, otherwise 0. A health centre that correctly did 75% of the activities was graded as good performance otherwise poor performance. Overall, the performance for each health centre level across the five financial years showed that poor performance in all the participating health centres, though control health centres performed better than the case health centres.” It doesn’t fit to the results section. The placement of the paragraph is not appropriate. Authors are advised to revise the placement.

Results

The first paragraph of the result section doesn’t seem understandable for readers. It seems confusing. Authors are advised to present it as concise and understandable to readers. It would be good if the narrative description of the first paragraph is about the performance of the health facilities based on the standard than comparing the health facilities of cases and controls. Better to compare the health facilities of cases and controls in next paragraphs.

On the final paragraph under the section: Districts' performance in annual health work planning in Busoga sub-region: authors stated: “Generally, the control facilities performed better than the control facilities in the subsequent five years”. There is an error on the statement, authors are requested to make a correction.

The results of the authors need to describe the performance of the health facilities based on the annual health work planning variables. Authors are advised to provide descriptions on the work plan of processes of the major health service indicators.

The result section would have been good if it is aligned with the methods section under the sub-heading of, a) District performance in annual health work planning, authors stated “In Uganda, ministry of health has set performance standards in line with maternal child health, health education and promotion, environmental determinants of health, water and sanitation, mental health and health-seeking behavior, amongst others for the lower health centre leaders.” From this paragraph, I would expect the set of performance standards to be indicated in the result section of this manuscript. However, the result of annual health work planning in this manuscript was not integrated with the performance indicators such as: maternal and child health, health education and promotion, environmental health, water and sanitation, mental health and health-seeking behavior. Thus, authors are requested to provide brief descriptions on the strength and weakness of the categories of health service units (Maternal and child health, water and sanitation … etc) of the annual health work planning.

On the result section of Competences of leaders of lower community health centres in annual health work planning authors stated: “For the 5 years, 80 percent of the health centres in the control and 83.3 percent in the case had not attended 4 or more planning cycle meetings organised by the district in the five years. Equally, most health centres did not organise annual health work planning meetings at the facility level, and less than 4 people were involved in the planning process of the work plan (43.3% for the controls and 70% for the cases).” How do authors rely to measure competencies of leaders in the aforementioned variables? It would be difficult to consider availability of staff and organizing meetings as measurements of leaders competency. Instead, authors are advised to use a combination of variables “staff being available all the time and staff being competent, productive and responsive” according the recommendation of World Health Organization (WHO).

Table IV. Performance of the health facilities in Busoga sub-region across 5 fiscal years period:

The table shows only percentages; authors are requested to include the frequencies beside the percentages for clarity.

Zero count is not clear, so authors are advised to provide brief information.

The HCIV performance in table IV is reported in FY 2021/22 is reported 9.1% while in 2020/2021 it was 45.5%, authors need to provide the possible and potential reasons for the highly declining performance of this particular institution. Since evidence generation is useful for policy making decisions, authors should recommend possible solutions for this particular institution.

Table 5: Showing the competence of the health facilities

There a HUMC

Yes

No

How do authors decide to measure competency of the health facilities with one-time availability of staff? Authors didn’t show the five years’ data on table 5. Authors are advised to clearly indicate the five-year data in terms of health facility management committee availability. The table needs to be again developed in a way to show the five years data.

Staffing level meets the standards for the facility

Yes

No

Authors didn’t describe about the staffing level of competency to meet the standards of the facility. Authors are advised to indicate either standard or operational definitions that describe the staffing level to meet the standard of the facility.

Except one variable stated as “Tool used for developing annual health work plan” in table 5, authors didn’t illustrate a specific table related to availability of manuals, guidelines and standard operating procedures that helps to staff members in performing effectively. Authors seem to overlook manuals and guidelines in improving the performance and competency of health workers in the health facilities. So, authors are requested to provide their reflections in this regard.

Table 6: Influence of competence of the lower health centre leaders in annual health work planning on the performance in the Busoga sub-region 2017/18-2021/22

Authors tried to calculate the Chi square and P value to see the competency of low health center leaders.

Due to the nature of susceptibility to selection bias, retrospective case–control studies need to be carefully handled using the best data analysis method that fits to the type of data collected. I am surprised why authors didn’t raise these issues as concern. The way the calculation is made is not clear. It would be difficult to understand if the frequency of each variable is not indicated on the table along the other column items. So, authors are advised to revise the statistical test, calculating odds ratio might be also another option, it can also show the confidence interval. Besides, the table should be reformulated consisting the frequency of each variable.

Discussion

This manuscript seems an evolving, I didn’t see similar manuscripts before. The discussion seems fine, but still needs further work. If the authors tried to go back to their data and accordingly incorporate the comments provided on the results section I hope additional similarities and variations of the new results could emerge on the discussion section.

Authors mentioned very important terminologies related to leaders competency in the middle of theeir manuscript such as: productivity, efficiency, effectiveness, and quality. Why do authors brought these terminologies in their manuscript, while not indicated in their findings? Could authors recheck their data if it captures more information?

Conclusion: the conclusion seems overriding the findings of the study. Authors stated “Health centres that meet staffing standards, actively participate in district planning cycles, and utilize a diverse set of implementation manuals for work planning are better positioned to contribute to district-wide health improvements.” Though the statement is true, but the highlighted phrase “staffing standards” was not appropriately defined and not comprehensively applied in the data collection tool. The data collection tool lacks comprehensiveness. So authors need to balance the intensity of the statement on the conclusion towards the actual findings of the data. Otherwise, it would be confusing for readers.

Recommendation

Authors need to check the alignment of the recommendations with their findings.

Decision: Major revision

Dr. Tesfay Gebregzeabher Gebrehiwet

Mekelle University, Ethiopia

**Do you want your identity to be public for this peer review?** For information about this choice, including consent withdrawal, please see our Privacy Policy

Reviewer #1: **Yes: ** kirya musa

Reviewer #2: **Yes: ** Dr. Tesfay Gebregzeabher Gebrehiwet

---

## [Author Response · Author response to Decision Letter 1]

26 Mar 2025

Reviewer#1: The word competence being a noun, would sound proper if it was ‘’COMPETENCES IN ANNUAL WORK PLANNING’’ than ‘’COMPETENCE IN WORK PLANS’’.

Response: The word competencies is a better one to use in this case (academic writing).

Reviewer #2: Title: is too long not catchy. Authors are advised to shorten their title to be attractive for readers.

If authors agree they can use the alternative title that reads as: “Competencies of frontline health facility leaders in annual work plan and its influence on district health management in Busoga Sub-Region: A Retrospective Study.”

Response: Competencies of frontline health facility leaders in annual work plan is catchy. However, it will consider leaders in all levels of health centres, yet this study looked at lower level (Centre IV and below) leaders where annual health work planning was not or partially done.

The title has been slightly changed and in line with the suggested ideas by the reviewer as;

“Competencies of Lower-Level Community Health Centre Leaders in Annual Health Work Planning and Their Influence on District Performance in Busoga Sub-Region: A Retrospective Study”

Reviewer #2: Background: Authors need to incorporate the essential gap that makes them to conduct this research study.

Response: The gap has been incorporated.

Reviewer #2: Objective: The first statement is adequate for the objective, while authors need to move the next statement indicated in the objective to the method’s section.

Response: Next statement was deleted under objective and transferred to methodology.

Reviewer #2: Methods: what is the justification of authors on the adequacy of the sample of health centers for cases and controls? Authors are advised to provide valid justification for the cutoff point “12”.

Authors need to briefly describe the techniques of data collection

Response: The sample of health centres for cases and controls are adequate in reference to previous studies like Muwanguzi, et al. (2020) study on the effectiveness of training of health unit management committee (HUMC) members, who quoted Schelssman’s (1982) formula to determine the sample size of health centres. Also, geographically, health centres in the region are scattered and resource constraints has to be keenly considered.

Techniques of data collection were structured interviews of leaders by research assistants (at least 5 in each centre) and documentary review (especially the annual work plan) for reviewing certain indicators suggested by the study, but also to verify answers provided by respondents.

Reviewer #2: Conclusion: authors are advised to make conclusion based on their findings.

Keywords: it is better if authors limit the key words to 5

Response: Conclusion has been modified according to findings.

Key words have been reduced to 5.

Reviewer #2: What is the major gap that needs to be explored by authors? Is it about the leadership competency of low health center leaders or organizational annual work plan?

Response: There is still a gap in terms of competence in making annual work plans among lower facilities leaders towards making effective annual work plans. This has been reflected in the main document

Reviewer #2: Authors are advised to clearly describe the pertinent components of:

1.The regional (African) experience of low health centers leader’s competency in developing annual work plan and its contribution on the district performance improvement

2.The flow of their description needs to be from global to local and from general to specific.

Response: Some studies from Ghana; Malawi and Uganda have been cited and included.

However, there’s no much literature on this exact subject matter.

Response: This has been addressed and the health system structure included.

Reviewer #2: Objectives of the study

Specific objectives: the specific objectives are not aligned with the broad objective.

Authors described the specific objectives: 1) To establish the districts' performance in annual work planning in the Busoga sub-region.

2) To establish the influence of competence of the lower community health centre leaders in

annual health work planning on the district performance in the Busoga sub-region.

Authors are advised to align the specific objectives with the general objectives.

Response: The objectives have been re phrased as;

To assess the level of performance of districts in annual health work planning in Busoga region.

To establish the competencies of lower level community health centre leaders in annual work planning that influence district performance in Busoga region.

Reviewer #1. Document/ records review checklist, Validity of instruments, Reliability of the instruments and Data analysis,

Use the right tense (past) and grammar in this section.

Response: The tenses have been changed to past tenses and the grammar corrected.

Reviewer #1. Dependent variable: The study considers the competences of the leaders of the health centers but in actual sense, it is the competences of the health facilities. The researcher may need to explicitly state the assumption that the health facility competences are synonymous with the leaders’ competences.

Response: The study considered competencies of lower level community health centre leaders in the health centres.

The dependent variable for this study is district performance.

Reviewer #1. Methodology: This is a case control study according to the researcher. The controls however do not meet the criteria set out by the study. In fact, the control group participants are cases by definition. There has to be a justification for this decision by the researcher or else this is a major methodological flaw.

It would have been better if the researchers considered a retrospective cohort of under performers (more so after noticing that the control group criteria was out of reach). In this case an independent t-test would have been a good choice at analysis.

Responses: This study considered performance of districts in order to come up with the two groups as cases for poor performance and controls for good performance. There was no intervention(s) in both the case and control groups. It is one reason the study investigated whether competencies of lower-level community health center leaders in annual health work planning influenced district performance in both groups.

Perhaps, this study can inform in future for a cohort study with a defined intervention to be followed.

In this study, the controls were districts with performance above the national average (NA) referred to as good performing districts for at least four years in the period 2017/18 to 2021/22 and these included Jinja, Iganga, Kamuli and Kaliro (see table 1). Then, the cases were districts with performance below the national average (NA) referred to as poor performing districts for at least three years in the period 2017/18 and included Namutumba, Namayingo, Mayuge and Luuka.

Linking competencies of leaders to the district performance, Odds Ratio quantifies how much likely are competent leaders perform better compared to incompetent leaders. This is well stated in the methodology under data analysis.

Reviewer #1. Conclusion: The study makes bold conclusions on the backdrop of a methodological error.

Responses: Methodological error has been corrected and clarified.

Reviewer #1. Readability: The manuscript needs to be trimmed to a more concise version.

Responses: The manuscript has been trimmed.

Reviewer #2. Authors are supposed to describe the number of facilities participated in the study as cases and controls.

Response: Description of facilities has been done.

Reviewer #2. In the methods section, Authors noticed: “There was a comparison of performance between the worst performing (case) and best performing (control) districts in the region according to the Annual Health Sector Performance reports from 2017/18 financial year to 2021/2022”. While in their result section they reported “the district performance in annual health work planning was not good in both the case and control groups (26.4% and 47.2% respectively).” How do authors justify for the 1.8 times fold of performance of control groups?

Responses: The poor performing districts and good performing districts were selected based on the national average score which the study based on to categorize the districts into cases and controls in the region. However, this did not guarantee how the two groups (cases and controls) can perform in terms of competences of the lower-level health centre leaders.

The average score for the priority areas was considered and 75% and above score was for good performing districts. However, averagely, the districts as cases and controls did not meet 75%, hence poor performance in all groups.

Reviewer #2. The result showed the 1.8 fold of best performing districts (controls) were with poor performance compared to the worst performing districts. How do authors justify the discrepancy that seems unrealistic?

Responses: That is what we discovered with our findings, that competencies of leaders did not influence the good performance of districts.

Reviewer #2. Authors stated “The study was conducted in the Busoga sub-regions of eleven (11) districts and one (1) city”, what does districts mean? Urban sub-urban or rural districts?

Response: Districts mean rural districts. It has been modified in the methodology.

Reviewer #2. Do the performance assessment standard fits to evaluate Jinja City equally with the other district? Do authors have reasons to include the City with the other district, doesn’t it create a bias to categorize city with other districts in research? For obvious reasons in cities, the competence of health workers and its managerial status is much better compared to the level of districts. Any reflections from the authors.

Responses:Constitutionally, in Uganda, a city is equivalent to a district. Therefore, in terms of stating districts making up Busoga region, we could not miss Jinja city.

However, the performance standard fits in the rural districts than the urban district. Jinja city being an urban would not meet the selection criteria.

Reviewer #2. On another note, authors are advised to provide clear picture of the study setting related to the structure of the health system, availability number of health workers and the access of health institutions to the population.

Responses: It has been addressed in the introduction and methodology section.

Reviewer #2. Authors used District performance and Competence in annual health work planning to measure the variables for performance. It seems many variables are measured in order to determine the status the outcome variable. How did authors manage the analysis of multiple variables?

Responses: The authors aggregated multiple indicators into a single performance score by assigning weights to each indicator based on its importance. This was done through summation and averaging. District performance was measured as a good one when three quarters (75%) of the priority areas identified in the facility work plan were realized. It was guided by the national average score which ranged from approximately 65 to 74 percent (approx. 3/4 of the performance mark.

Reviewer #2. Authors are advised to provide brief description about the type of health facilities in the study setting section. It is not clear for readers what the meaning of health center IV, III and II. The reason why authors used the stratified sampling technique is not clearly described in the manuscript.

Responses: It has been handled in the introduction and methodology.

Reviewer #2. Under the Document/Records Review Checklist sub heading, on the data collection section. Authors noted “A checklist will be a tool designed to capture information extracted from the annual health work…….” The “will be” phrase is not acceptable at this stage; authors are advised to us the appropriate form of tense. Similar error is shown on the statement provided under data analysis stated as “The pvalue set at 0.05 will be used to determine the statistical significance ……… at 95 percent confidence intervals (CI)”. “It will derive regression odds ratios and p-values” Authors are advised to make correction on the highlighted phrases. Correction form of the tense is needed.

Responses: Tenses have been corrected

Reviewer #2. The sub heading stated by authors “Districts' performance in annual health work planning in Busoga sub-region” and its content in the first paragraph doesn’t fit to the results section. Authors need to consider to move it to the methods section, because it is a descriptive narrative about the performance assessment and how health centers were graded based on their performance.

Responses: Paragraph shifted to methodology .

Reviewer #2. Similarly, the second paragraph that reads as “Each correct activity was scored 1, otherwise 0. A health centre that correctly did 75% of the activities was graded as good performance otherwise poor performance. Overall, the performance for each health centre level across the five financial years showed that poor performance in all the participating health centres, though control health centres performed better than the case health centres.” It doesn’t fit to the results section. The placement of the paragraph is not appropriate. Authors are advised to revise the placement.

Responses: The paragraph has been rephrased and shifted to methodology'

Reviewer #2. The first paragraph of the result section doesn’t seem understandable for readers. It seems confusing. Authors are advised to present it as concise and understandable to readers. It would be good if the narrative description of the first paragraph is about the performance of the health facilities based on the standard than comparing the health facilities of cases and controls. Better to compare the health facilities of cases and controls in next paragraphs.

Responses: This has been done.

Reviewer #2. On the final paragraph under the section: Districts' performance in annual health work planning in Busoga sub-region: authors stated: “Generally, the control facilities performed better than the control facilities in the subsequent five years”. There is an error on the statement, authors are requested to make a correction.

Responses: Error has been corrected.

Reviewer #2. The results of the authors need to describe the performance of the health facilities based on the annual health work planning variables. Authors are advised to provide descriptions on the work plan of processes of the major health service indicators.

Response: This has been corrected.

Reviewer #2. The result section would have been good if it is aligned with the methods section under the sub-heading of, a) District performance in annual health work planning, authors stated “In Uganda, ministry of health has set performance standards in line with maternal child health, health education and promotion, environmental determinants of health, water and sanitation, mental health and health-seeking behavior, amongst others for the lower health centre leaders.” From this paragraph, I would expect the set of performance standards to be indicated in the result section of this manuscript. However, the result of annual health work planning in this manuscript was not integrated with the performance indicators such as: maternal and child health, health education and promotion, environmental health, water and sanitation, mental health and health-seeking behavior. Thus, authors are requested to provide brief descriptions on the strength and weakness of the categories of health service units (Maternal and child health, water and sanitation … etc) of the annual health work planning.

Responses: This has been corrected.

This has been summarized into good and poor performance of districts in the control and cases as districts respectively. Variables under performance of the district have guided in understanding the general performance of districts in annual health work planning.

Reviewer #2. On the result section of Competences of leaders of lower community health centres in annual health work planning authors stated: “For the 5 years, 80 percent

---

## [Decision Letter · Decision Letter 1]

Dear Dr. Muluya,

Thank you for submitting your manuscript to PLOS ONE. After careful consideration, we feel that it has merit but does not fully meet PLOS ONE’s publication criteria as it currently stands. Therefore, we invite you to submit a revised version of the manuscript that addresses the points raised during the review process.

**ACADEMIC EDITOR: The revised paper has improved and could be publishable. Please resolve the outstanding observation from one of the reviewers. In addition, proof-read the paper to reduce on language mistakes.**

We look forward to receiving your revised manuscript.

Kind regards,

Aloysius Gonzaga Mubuuke

Academic Editor

PLOS ONE

Journal Requirements:

Reviewers' comments:

Reviewer's Responses to Questions

**Comments to the Author**

Reviewer #1: (No Response)

Reviewer #2: All comments have been addressed

2. Is the manuscript technically sound, and do the data support the conclusions?

Reviewer #1: Yes

Reviewer #2: Yes

3. Has the statistical analysis been performed appropriately and rigorously?

Reviewer #1: Yes

Reviewer #2: Yes

4. Have the authors made all data underlying the findings in their manuscript fully available?

Reviewer #1: Yes

Reviewer #2: Yes

5. Is the manuscript presented in an intelligible fashion and written in standard English?

Reviewer #1: Yes

Reviewer #2: Yes

Reviewer #1: The authors have satisfactorily addressed most of the comments.

However, one issue is still unresolved. the authors refer to this as a case control study in their methods and results. The authors clearly defined their criteria for cases and controls using a national average of 75% in performance; implying that the scores above 75% good performance( controls) and the performance scores below 75% poor performance( cases).

The authors admit that they didn't get the outcome that meets the above definition. In fact all the sampled population fall under the category of cases and therefore doesn't give a good comparison for this study. "Overall, the performance of districts in annual health work planning was poor in both cases and controls, at 26.4% and 47.2%, respectively". If this statement is correct then this ceases to be a case-control study. I would suggest that the authors simply call this a retrospective cohort and the regression analysis done still conveys the same message. Otherwise, this would be misleading to the reading community.

Once the above have been addressed, the paper should be ready for publication.

Reviewer #2: Dear Editor,

I hereby submit revised report of the manuscript, entitled “Competencies of Lower-Level Community Health Centre Leaders in Annual Health Work Planning and Their Influence on District Performance in Busoga Sub Region: A Retrospective Study”, after reading and checking all the revised manuscript. I acknowledge the efforts of the authors for revising the manuscript and developed a good evidence which is highly needed. Now, authors have incorporated all comments and I can say the manuscript is now in better position. I suggested to be accepted immediate after correction made for the underneath comments by authors. I believe, this manuscript will be very helpful in forwarding better evidence for decision making in rebuilding the human resource development and improving the performance of health centers.

Revised title: “Competencies of Lower-Level Community Health Centre Leaders in Annual Health Work Planning and Their Influence on District Performance in Busoga Sub Region: A Retrospective Study”

Authors need to revise the comments below.

Authors wrote the specific objectives with a verb that is not explanatory. “To establish”

Specific objectives 1) To establish the districts' performance in annual work planning in the Busoga sub-region. 2) To establish the influence of competence of the lower community health centre leaders in annual health work planning on the district performance in the Busoga sub-region.

The phrase needs to be replaced with an action verb. “To assess” or with another action verb that fits with the objective statement.

Dr. Tesfay Gebregzabher Gebrehiwet

Associate Professor of Public Health, Mekelle University, College of Health Sciences

**Do you want your identity to be public for this peer review?** For information about this choice, including consent withdrawal, please see our Privacy Policy

Reviewer #1: **Yes: ** Kirya Musa

Reviewer #2: **Yes: ** Dr. Tesfay Gebregzabher Gebrehiwet

---

## [Author Response · Author response to Decision Letter 2]

23 Apr 2025

Reviewer #1: The authors have satisfactorily addressed most of the comments.

However, one issue is still unresolved. The authors refer to this as a case control study in their methods and results. The authors clearly defined their criteria for cases and controls using a national average of 75% in performance; implying that the scores above 75% good performance (controls) and the performance scores below 75% poor performance (cases).

The authors admit that they didn't get the outcome that meets the above definition. In fact all the sampled population fall under the category of cases and therefore doesn't give a good comparison for this study. "Overall, the performance of districts in annual health work planning was poor in both cases and controls, at 26.4% and 47.2%, respectively". If this statement is correct then this ceases to be a case-control study. I would suggest that the authors simply call this a retrospective cohort and the regression analysis done still conveys the same message. Otherwise, this would be misleading to the reading community.

Once the above have been addressed, the paper should be ready for publication.

Response:

As authors, we praise the reviewers for appreciating our efforts and keenly guiding us to fine-tune our work.

Our original design intent is a case-control study and a retrospective case-control study, which is defined as; cases – districts that performed poorly according to the national average score in annual health sector performance report for the past five years and controls – districts that performed well (above national average).

The authors believed something (exposure) must have contributed to the better or poor performance of the districts in the past years which perhaps was not reported in the national health sector performance report. That is when authors thought about competencies of health centre leaders in annual work plan development (as exposure) in the cases and controls. Therefore, the cases and controls had similar exposure (competencies of health centre leaders in work plan development). So, the exposure (competencies of leaders) doesn’t differentiate the groups.

The outcome was district performance in the annual health work plan development, which was low for the cases and controls, at 26.4% and 47.2%, respectively. This means both groups had weaknesses / limited competencies of leaders in annual health work plan development, but it does not change the fact that they were originally categorized by overall district performance (as per the annual health sector performance reports).

A null finding is still valid and useful — it tells us that something other than leader competence in annual health work plan development likely explains district performance differences.

Shifting from a case-control to a cohort study is definitely possible, but it requires reframing the study design, group selection, and sometimes data structure. For example it will require selecting districts based on exposure (competence level of health center leaders in annual work plan development). That is, districts with competent health center leaders will be the exposed cohort and those with incompetent ones as unexposed cohort. Then, next will be to assess district performance. We shall be interested in knowing if districts with competent leaders perform better over time compared to those with less competent ones.

Even for a retrospective study, cohort study will require to start with exposure (that is, competent leaders vs. incompetent leaders) and followed up in the groups of exposed cohort and unexposed cohort respectively to see what outcomes arise.

The case-control study did not compare competent and incompetent leaders in annual health work plan development.

Lastly, analysis is most likely to change. Retrospective cohort study may change to Risk Ratios instead of Odds ratios for the case-control.

If the explanation above is convincing, we request the study remains a valid retrospective case-control study despite of the finding that the hypothesized exposure didn’t explain the observed differences. It’s still important evidence.

Otherwise a cohort study will require major changes.

Reviewer #2: Authors wrote the specific objectives with a verb that is not explanatory. “To establish”

Specific objectives 1) To establish the districts' performance in annual work planning in the Busoga sub-region. 2) To establish the influence of competence of the lower community health centre leaders in annual health work planning on the district performance in the Busoga sub-region.

The phrase needs to be replaced with an action verb. “To assess” or with another action verb that fits with the objective statement.

Response:

The verb (establish) which is not explanatory was changed to active verb (assess) for both objectives as;

1) To assess the districts' performance in annual work planning in the Busoga sub-region.

2) To assess the influence of competence of the lower community health centre leaders in annual health work planning on the district performance in the Busoga sub-region.

---

## [Decision Letter · Decision Letter 2]

Competencies of lower-level community health centre leaders in annual health work planning and their influence on district performance in Busoga sub-region: A retrospective study

PONE-D-24-52873R2

Dear Dr. Muluya,

We’re pleased to inform you that your manuscript has been judged scientifically suitable for publication and will be formally accepted for publication once it meets all outstanding technical requirements.

Kind regards,

Aloysius Gonzaga Mubuuke

Academic Editor

PLOS ONE

Additional Editor Comments (optional):

No more comments

Reviewers' comments:

Reviewer's Responses to Questions

**Comments to the Author**

Reviewer #1: All comments have been addressed

2. Is the manuscript technically sound, and do the data support the conclusions?

Reviewer #1: Yes

3. Has the statistical analysis been performed appropriately and rigorously?

Reviewer #1: Yes

4. Have the authors made all data underlying the findings in their manuscript fully available?

Reviewer #1: Yes

5. Is the manuscript presented in an intelligible fashion and written in standard English?

Reviewer #1: Yes

Reviewer #1: The authors have satisfactorily addressed my comments.

Please proceed with publishing this work, it provides very important health policy related insights.

**Do you want your identity to be public for this peer review?** For information about this choice, including consent withdrawal, please see our Privacy Policy

Reviewer #1: **Yes: ** Kirya Musa

---

## [Editor Report · Acceptance letter]

PONE-D-24-52873R2

PLOS ONE

Dear Dr. Muluya,

I'm pleased to inform you that your manuscript has been deemed suitable for publication in PLOS ONE. Congratulations! Your manuscript is now being handed over to our production team.

Kind regards,

on behalf of

Dr. Aloysius Gonzaga Mubuuke

Academic Editor

PLOS ONE